# Deep Polynomial Chaos Expansion

**Johannes Exenberger**[1,2]  **Sascha Ranftl**[*,3,4]  **Robert Peharz**[*,2,5]

[*]Shared supervision
[1]TU Wien, Austria
[2]TU Graz, Austria
[3]Brown University, US
[4] Courant Institute NYU, US
[5]Graz Center for Machine Learning, Austria

## Abstract

Polynomial chaos expansion (PCE) is a classical and widely used surrogate modeling technique in physical simulation and uncertainty quantification. By taking a linear combination of a set of basis polynomials—orthonormal with respect to the distribution of uncertain input parameters—PCE enables tractable inference of key statistical quantities, such as (conditional) means, variances, covariances, and Sobol sensitivity indices, which are essential for understanding the modeled system and identifying influential parameters and their interactions. As the number of basis functions grows exponentially with the number of parameters, PCE does not scale well to high-dimensional problems. We address this challenge by combining PCE with ideas from probabilistic circuits, resulting in the *deep polynomial chaos expansion* (DeepPCE)—a deep generalization of PCE that scales effectively to high-dimensional input spaces. DeepPCE achieves predictive performance comparable to that of multi-layer perceptrons (MLPs), while retaining PCE's ability to compute *exact* statistical inferences via simple forward passes.

## 1 INTRODUCTION

Numerical simulations for partial differential equations (PDEs) are key tools in many areas of the physical sciences. The accuracy of PDE solvers typically depends on several uncertain input parameters, including boundary conditions, initial states, and physical properties. A common method to estimate these parameters is to train a *surrogate model* on a limited number of simulations, enabling efficient exploration of the parameter space. A widely used surrogate model is the *Polynomial chaos expansion* (PCE) [Wiener, 1938, Xiu and Karniadakis, 2002].

PCE models a function as a linear combination of polynomial basis functions that are orthonormal with respect to the input distribution. This orthonormality offers both theoretical and computational advantages. In theory, PCE yields the optimal polynomial approximation in a least-squares sense and, in the context of stochastic differential equations, allows coefficient computation similar to Galerkin projection [Galerkin, 1915, Ghanem and Spanos, 1991]. In practice, PCE enhances data efficiency and numerical stability, and statistical quantities of PCEs such as means, variances, covariances, and Sobol sensitivity indices [Sobol', 1990] have closed-form expressions [Sudret, 2008, Crestaux et al., 2009]. However, PCE scales poorly to high-dimensional parameter spaces, as the number of polynomial terms grows combinatorially with the number of inputs. Neural networks are thus increasingly used as surrogate models [Raissi et al., 2019, Karniadakis et al., 2021], as they often achieve better data fit for complex, high-dimensional physical systems. One major drawback of neural surrogates is that expectations and Sobol indices cannot be computed exactly, and approximations via Monte Carlo is often expensive and imprecise.

We propose a scalable and principled extension of PCE to high-dimensional inputs by combining it with a deep circuit architecture inspired by the framework of *probabilistic circuits* [Choi et al., 2020], which generalize shallow mixture models such as Gaussian mixture models (GMMs) into deep and structured representations. Like PCE, GMMs and other mixture models suffer from the curse of dimensionality. Deep probabilistic circuits overcome this problem by compactly representing exponentially many mixture components through circuit depth, enabling effective modeling of high-dimensional distributions [Peharz et al., 2020b,a]. We apply the same principle to introduce *deep polynomial chaos expansion* (DeepPCE), which can represent exponentially many orthogonal polynomial terms compactly. We derive exact formulas for statistical moments (means, variances, covariances) and Sobol sensitivity indices within the DeepPCE framework and show in experiments that DeepPCE (i) en-

*Accepted for the 8th Workshop on Tractable Probabilistic Modeling at UAI  (TPM 2025).*

ables scalable sensitivity analysis on a 100-dimensional synthetic function with analytic Sobol indices, and (ii) serves as a scalable surrogate for high-dimensional PDE benchmarks (Darcy flow, steady-state diffusion), matching the predictive performance of MLPs.

## 2 BACKGROUND

We denote random variables by uppercase letters (e.g. $X$, $Y$, $Z$) and corresponding values by lowercase letters (e.g. $x$, $y$, $z$). The set of integers $\{0, \ldots, K\}$ is denoted by $[K]$.

**Polynomial Chaos Expansion (PCE)** Numerical methods for modeling physical systems are often governed by a large number of uncertain input parameters, such as boundary and initial conditions or material properties. We denote these parameters by $\mathbf{X} = \{X_1, X_2, \ldots, X_D\}$, forming a $D$-dimensional random vector following a factorized distribution $p(\mathbf{X}) = \prod_d p(X_d)$. A PCE approximates the *response function* $f^*(\mathbf{X})$, i.e., the output of the numerical simulation, as a polynomial expansion

$$f_{\text{PCE}}(\mathbf{x}) = \sum_{\boldsymbol{\alpha} \in \mathcal{A}} w_{\boldsymbol{\alpha}} \boldsymbol{\Phi}_{\boldsymbol{\alpha}}(\mathbf{x}), \qquad (1)$$

where $\boldsymbol{\alpha} = (\alpha_1, \ldots, \alpha_D) \in \mathcal{A}$ is a multi-index, each $\alpha_d \in [K]$ denoting the polynomial degree associated with $X_d$ where $K$ is the maximum order of the expansion, and $\mathcal{A}$ is a set of multi-indices. $\boldsymbol{\Phi}$ are multivariate basis functions constructed as tensor products of univariate polynomials $\phi$:

$$\boldsymbol{\Phi}_{\boldsymbol{\alpha}}(\mathbf{x}) = \prod_{d=1}^{D} \phi_{\alpha_d}(x_d). \qquad (2)$$

Crucially, the univariate polynomials $\phi_{\alpha_d}$ are chosen to be orthonormal with respect to the distribution of $X_d$. Because we assume a factorized input distribution $p(\mathbf{X})$, the tensor-product polynomials are also orthonormal, yielding

$$\mathbb{E}_{\mathbf{X}}[\boldsymbol{\Phi}_{\boldsymbol{\alpha}}(\mathbf{X}) \boldsymbol{\Phi}_{\boldsymbol{\alpha}'}(\mathbf{X})] = \int p(\mathbf{x}) \boldsymbol{\Phi}_{\boldsymbol{\alpha}}(\mathbf{x}) \boldsymbol{\Phi}_{\boldsymbol{\alpha}'}(\mathbf{x}) \mathrm{d}\mathbf{x} = \delta_{\boldsymbol{\alpha}, \boldsymbol{\alpha}'}$$
$$(3)$$

where $\delta_{\boldsymbol{\alpha}, \boldsymbol{\alpha}'}$ is the Kronecker delta.

The number of possible basis functions is bounded by $|\mathcal{A}| \leq \frac{(K+D)!}{K!D!}$, which grows combinatorially with the number of inputs $D$ and the polynomial order $K$, quickly becoming infeasible with increasing dimensionality. Truncation schemes [Mühlpfordt et al., 2017] and sparse adaptive expansion methods [Blatman and Sudret, 2011] are often applied to improve scalability, still PCE typically scales to only a few dozen input parameters.

**Sensitivity Analysis** PCE has closed-form solutions for the expectation and variance of $f$. Since the zeroth-order polynomial, corresponding to $\boldsymbol{\alpha}_0 = (0, \ldots, 0)$, is a constant,

it follows from (3) that $\mathbb{E}[\boldsymbol{\Phi}_{\boldsymbol{\alpha}}(\mathbf{X})] = 0 \,\forall \boldsymbol{\alpha} \neq \boldsymbol{\alpha}_0$, hence

$$\mathbb{E}[f(\mathbf{X})] = w_{\boldsymbol{\alpha}_0}. \qquad (4)$$

Due to (3), all multiplicative cross-terms cancel under in expectation and we get

$$\text{Var}[f(\mathbf{X})] = \mathbb{E}[f_{\text{PCE}}^2(\mathbf{X}) - w_{\boldsymbol{\alpha}_0}] = \sum_{\boldsymbol{\alpha} \in \mathcal{A} \setminus \{\boldsymbol{\alpha}_0\}} w_{\boldsymbol{\alpha}}^2. \quad (5)$$

PCE also allows to compute "advanced" statistical quantities, such as variances of conditional expectations: Let $\mathcal{I} \subseteq \{1, \ldots, D\}$ be an index set for some parameters in $\mathbf{X}$ and $\neg \mathcal{I} = \{1, \ldots, D\} \setminus \mathcal{I}$. It can be shown that

$$\text{Var}_{\mathbf{X}_{\mathcal{I}}}[\mathbb{E}_{\mathbf{X}_{\neg \mathcal{I}}}[f(\mathbf{X}) \mid \mathbf{X}_{\mathcal{I}}]] = \sum_{\boldsymbol{\alpha} \in \mathcal{A}_{\mathcal{I}}} w_{\boldsymbol{\alpha}}^2, \qquad (6)$$

where $\mathcal{A}_{\mathcal{I}} = \{\boldsymbol{\alpha} \colon \forall j \in \neg \mathcal{I} \colon \alpha_j = 0\}$. These variances of conditional expectations allows to compute prominent sensitivity measures such as *Sobol indices* [Sobol', 1990]. These indices are normalized variances of the so-called *Sobol decomposition*, which describes $f$ as a superposition of all possible $2^D$ interaction terms Saltelli et al. [2008], including a constant, $D$ individual terms, $\binom{D}{2}$ pair-wise terms, etc.

The *first-order Sobol indices* $S_i$ describe the direct contribution of each parameter $X_i$ to the variance of the response function. In PCE it is given via (5) and (6) as

$$S_i = \frac{\text{Var}_{X_i}(\mathbb{E}_{\neg \mathbf{x}_i}[f(\mathbf{X}) \mid X_i])}{\text{Var}[f(\mathbf{X})]}. \qquad (7)$$

Other Sobol indices, describing for example pairwise interactions, can also be computed in PCE.

**Probabilistic Circuits** Probabilistic circuits [Choi et al., 2020] are computational graphs with structural constraints that enable tractable inference. Although primarily used for probabilistic modeling, they can be applied to regression tasks, in which case they are typically referred to as (structured) circuits or sum-product networks [Poon and Domingos, 2012]. Formally, a circuit $\mathcal{C}$ is a parameterized directed acyclic graph defining a function $\mathcal{C}(\mathbf{X})$ with input variables $\mathbf{X}$. Let $\text{ch}(c)$ be the set of children nodes of some node $c \in \mathcal{C}$. There are three types of circuit nodes. An input node $c$ (i.e. $\text{ch}(c) = \emptyset$) represents a parameterized function $g_c(\mathbf{X}_c)$ over a subset of inputs $\mathbf{X}_c \subseteq \mathbf{X}$, called its *scope*. Internal nodes (i.e. $\text{ch}(c) \neq \emptyset$) are either product nodes or sum nodes, representing operations $\prod_{c' \in \text{ch}(c)} f_{c'}(\mathbf{X}_{c'})$ and $\sum_{c' \in \text{ch}(c)} w_{c,c'} f_{c'}(\mathbf{X}_{c'})$, respectively, where $w_{c,c'}$ are the sum node parameters. The scope of a sum or product node is recursively given as $\mathbf{X}_c = \bigcup_{c' \in \text{ch}(c)} \mathbf{X}_{c'}$. To enable tractable inference, circuits have to be *smooth* and *decomposable*. A circuit is smooth when the children of every sum node have identical scope. Formally, $\mathbf{X}_{c'} = \mathbf{X}_{c''} \,\forall c', c'' \in \text{ch}(c) \,\forall c \in \Sigma$, where $\Sigma$ is

the set of sum nodes. A circuit is decomposable when the children of every product node have pairwise disjoint scope, $\mathbf{X}_{c'} \cap \mathbf{X}_{c''} = \emptyset \ \forall c', c'' \in \text{ch}(c) \ \forall c \in \Pi$, where $\Pi$ is the set of product nodes.

# 3 DEEP POLYNOMIAL CHAOS

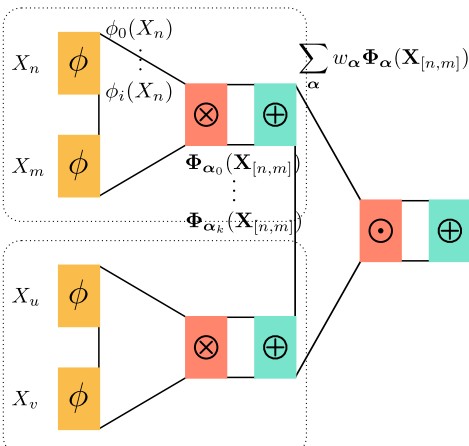

Figure 1: A DeepPCE with two PCE input nodes over scopes $\{X_n, X_m\}$ and $\{X_u, X_v\}$. Inputs are expanded as orthogonal polynomials $\phi_i$. A layer of outer-product nodes $\otimes$ forms the multivariate basis functions (tensor-products). A sum layer $\oplus$ composes a PCE with trainable network weights, followed by one or several blocks of element-wise products $\odot$ and weighted sums $\oplus$ over these products.

The core idea is to define the input nodes in probabilistic circuits using orthonormal PCE basis functions (see Figure 1). Circuits can be interpreted as representations of polynomials [Choi et al., 2020, Vergari et al., 2021], and conversely, a PCE can naturally be viewed as a circuit. The integrals for the expectation $\mathbb{E}[f(\mathbf{X})]$ and the variance $\text{Var}(f(\mathbf{X}))$ in PCEs can be computed very efficiently even for high dimensional input spaces (4, 5). Constructing a PCE via hierarchical sums and products of PCEs with smaller scopes enables the modeling of high-dimensional functions beyond the reach of traditional PCE approaches. We implement the circuit as a layerwise tensorized computational graph Loconte et al. [2024], following the structure of modern implementations Vergari et al. [2019], Peharz et al. [2020a]. We denote the multivariate circuit output as $\mathbf{Y} = \{Y_1, \dots, Y_O\}$, representing the function $\mathbf{Y} = f(\mathbf{X})$, where the input nodes are defined as PCEs.

**PCE layer** The first layer of the circuit consists of multiple input units, each associated with a scope $\mathbf{X}_c \subseteq \mathbf{X}$. In contrast to probabilistic circuits, where input nodes encode probability distributions, DeepPCE input nodes each represent a PCE, denoted as $g_c(\mathbf{X}_c)$ (see Eq. 1). We use overparameterized circuits Loconte et al. [2024], with multiple input nodes encoding a PCE $g_{c,n}(\mathbf{X}_c)$ over the same

scope, each with a distinct set of weights. In the special case where $\mathbf{X}_c = \mathbf{X}$, the entire input space is used as a single scope, recovering a shallow PCE. We adopt a random partitioning strategy for assigning scopes to the input nodes, similar to the RAT-SPN architecture [Peharz et al., 2020b]. However, rather than mixing multiple randomly partitioned circuits at the root node, we restrict the model to a single random partition of the input vector $\mathbf{X}$ to preserve circuit decomposability for tractable inference [Vergari et al., 2021].

**Sum-product layer** We abstract product and sum operations in one single module for computational efficiency as shown in Peharz et al. [2020a]. Each product unit computes the element-wise product from its input vectors $\mathbf{u}, \mathbf{v}$ with length $M$ and scopes $\mathbf{X}_{c'}$ and $\mathbf{X}_{c''}$ respectively, based on a Hadamard product layer [Loconte et al., 2024]. The scope of a product node output is the union of the scope of its inputs $\mathbf{X}_{c'} \cup \mathbf{X}_{c''}$, the output is $\mathbf{o}_n = \mathbf{u} \odot \mathbf{v}$. For each product node $p_n$, there exist $K \geq 1$ individual sum nodes, each computing a sum over the same scope induced by the input node $p_n$ but with a distinct set of weights. The output of a sum node is $s_{n,k} = \sum_m^M w_m^{k,n} o_m$, where $w_m^{k,n}$ are the elements of the sum units' weight vector $\mathbf{w}_{k,n} \in \mathbf{W}_{K \times N \times M}$ and $\mathbf{W}$ is the complete weight tensor of the sum layer. The number of sum nodes $K$ for each product node is a hyperparameter controlling the overparameterization of the circuit. At the output layer, $K = 1$ and each sum $s_n$ simply corresponds to one output dimension $Y_n \in \mathbf{Y}$. The depth of the DeepPCE is determined by the input dimensionality $D$ and the size of the input scopes $\mathbf{X}_c$.

**Inference** By building on both PCE and circuit properties, the DeepPCE retains the ability to compute exact statistical moments and sensitivity indices. We show how to compute the total model expectation $\mathbb{E}[f(\mathbf{X})]$ and covariances $\text{cov}(f(\mathbf{X}))$. Under the assumption that $p(\mathbf{X})$ factorizes, the expectation is represented by the same circuit as $f(\mathbf{X})$, following from the fact that (i) sums commute with expectations and (ii) factorized expectations distribute over decomposable products. The single-dimensional expectations can hence be "pulled down" to the input nodes [Choi et al., 2020, Vergari et al., 2021]. At the input nodes, the expectation $\mathbb{E}[g(\mathbf{X}_c)]$ of a PCE node with scope $\mathbf{X}_c$ can be estimated directly from the weights as shown in (4) by simply setting all weights $w_{\boldsymbol{\alpha}_i} = 0 \ \forall i > 0$. We get $\mathbb{E}[f(\mathbf{X})]$ by using $\mathbb{E}[g(\mathbf{X}_c)]$ and performing a standard forward pass through the circuit. Similarly, we get $\text{cov}(f(\mathbf{X}), f(\mathbf{X}))$ by computing $\text{cov}(g(\mathbf{X}_c), g(\mathbf{X}_c))$ at the input PCE — multiplicative cross-terms cancel due to orthogonality — according to 5, followed by a forward pass. Applying the same rationale, we are able compute other statistical quantities, such as conditional expectations, conditional covariances, expectations of conditional covariances and covariances of conditional expectations, as shown in Appendix A.

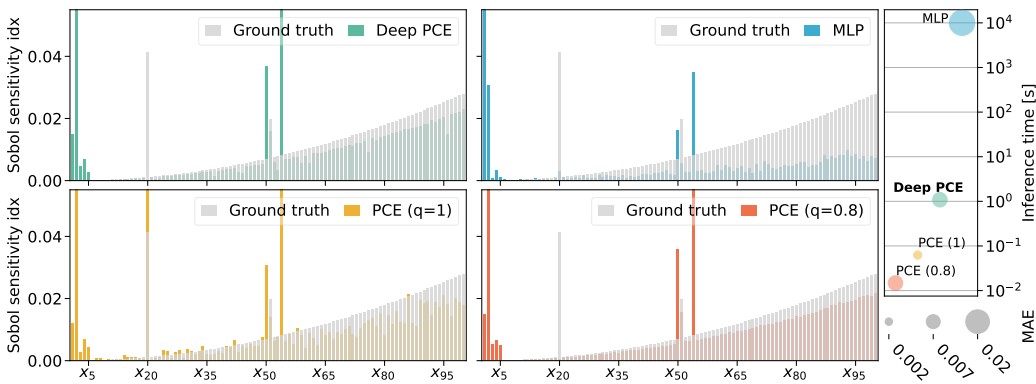

Figure 2: First order Sobol indices for the **PCE benchmark** dataset using 10000 data samples. Indices are normalized by their total sum for better comparability. Computation times (wall clock time) for Monte Carlo approximated Sobol indices for the MLP are larger by a factor of $10^4$ compared to the tractable operations with the DeepPCE.

## 4   EXPERIMENTS

We test our DeepPCE on a common PCE benchmark function and two high-dimensional PDE benchmarks and compare it to a standard MLP and different variants of a shallow PCE. The DeepPCE is domain-agnostic and similary applicable to different types of input data, hence we consider MLPs the most adequate baseline. We report best performing results for both MLP and DeepPCE based on multiple training runs. Relative MSEs are computed by $\frac{\frac{1}{N}\sum_i^N (y-\hat{y})^2}{\frac{1}{N}\sum_i^N y^2}$. Inputs for both two-dimensional PDE datasets are $32 \times 32$ dimensional fields — PCEs have so far not been applicable to such high-dimensional problems.

**PCE Benchmark**   The *100D function* $f : \mathbb{R}^{100} \mapsto \mathbb{R}$ is a common benchmark for sensitivity analysis with PCEs [Lüthen et al., 2021]. We compare the DeepPCE to two standard PCEs using different truncations, (i) a truncation based on the total order of the expansion ($\|\boldsymbol{\alpha}\|_1 \leq K$) ('PCE $q = 1$'), (ii) a PCE using a hyperbolic truncation scheme ($\|\boldsymbol{\alpha}\|_{0.8} \leq K$) ('PCE $q = 0.8$') [Blatman and Sudret, 2011] with $K = 3$. First order Sobol indices (7) are presented in Figure 2, showing that the DeepPCE performs on-par with the best shallow PCE in identifying variance contributions of single input variables. For the MLP, Sobol indices are approximated using Monte Carlo with a grid-based sampling approach. For each $X_i$, we use $10^9$ samples to compute $\mathrm{Var}_{X_i}(\mathbb{E}_{\neg \mathbf{X}_i}[Y \,|\, X_i])$. Measured wall clock time for computing MLP Sobol indices is longer by a factor of $10^4$ compared to the analytical computation with the DeepPCE (hours vs. a single second).

**Darcy flow**   This two-dimensional PDE is a common benchmark for data-driven physics models. We use the dataset provided by Zhu and Zabaras [2018]. The input is a random permeability field, the output is a velocity field. Training was performed using 5000 training samples, while 5000 data samples were used for validation and evalua-

tion. Another 2000 samples were used as holdout test set. The DeepPCE shows competitive predictive performance compared to a MLP, with relative MSEs of 0.0550 for the DeepPCE and 0.0489 for the MLP. Randomly selected test samples are shown in appendix D.

**Steady-state diffusion**   The two-dimensional PDE consists of a random diffusion coefficient modeled as a Gaussian random field and a corresponding pressure field as output. The data was generated using the solver provided by Tripathy and Bilionis [2018]. We generate a dataset with 10000 samples and a holdout test set with 2000 samples. As in the Darcy flow experiment, we train each network with 5000 samples and use the remaining 5000 samples for validation and evaluation. The DeepPCE shows predictive performance on par with the MLP on the holdout test set — both models manage to fit the data almost perfectly (relative MSEs of 0.0001 for the DeepPCE and 0.0002 for the MLP). Randomly selected test samples are shown in appendix D.

## 5   CONCLUSION

We introduce Deep Polynomial Chaos Expansion (Deep-PCE), a generalization of classical PCE that embeds orthonormal polynomial bases within a deep circuit architecture. DeepPCE enables exact computation of statistical moments and Sobol sensitivity indices while scaling to high-dimensional inputs, achieving predictive accuracy comparable to MLPs. DeepPCE supports fully analytical uncertainty quantification via simple forward passes and overcomes the curse of dimensionality of PCEs using hierarchical tensor products. Limitations include sensitivity to initialization and reliance on factorized input distributions, suggesting future work on robustness and correlated inputs. Overall, DeepPCE provides (i) a principled, fast mechanism for feature importance analysis and (ii) a scalable and tractable alternative for surrogate modeling in science and engineering.

## Acknowledgements

This work is part of the project VENTUS (910263), which has received funding from the FFG AI for Green program.

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

# Deep Polynomial Chaos Expansion
# (Supplementary Material)

**Johannes Exenberger**[1,2]  **Sascha Ranftl**[*,3,4]  **Robert Peharz**[*,2,5]

[*]Shared supervision
[1]TU Wien, Austria
[2]TU Graz, Austria
[3]Brown University, US
[4] Courant Institute NYU, US
[5]Graz Center for Machine Learning, Austria

## A  INFERENCE

### A.1  CIRCUIT PRELIMINARIES

Our goal is to compute expectations and covariances to perform sensitivity analysis and uncertainty quantification with the DeepPCE. The DeepPCE represents a function $f(\mathbf{X}) = \mathbf{Y}$, where $\mathbf{Y} = \{Y_1, \dots, Y_O\}$ is the $O$-dimensional vector of outputs and the input $\mathbf{X} = \{X_1, \dots, X_D\}$ is a $D$-dimensional vector of random variables, following a factorized distribution:

$$p(\mathbf{X}) = \prod_{d=1}^{D} p(X_d). \tag{8}$$

The DeepPCE shares its structure with (probabilistic) circuits — the constraints imposed on the structure of circuits allow to perform tractable inference. For detailed explanations and proofs of the theoretical foundations and necessary and sufficient conditions for tractability, we refer to Choi et al. [2020], Vergari et al. [2021], Loconte et al. [2024]. Here, we will provide a higher-level summary of the important concepts. Similar to other frameworks [Peharz et al., 2020a, Loconte et al., 2024], the DeepPCE is implemented as layerwise tensorized computational graph, parallelizing the operations inside the circuit. It still consists of the same basic computational *nodes*, which we explain briefly. An inner node (a node that is not an input node) in the circuit receives inputs from one or more other circuit nodes, its *children*. We denote the children of a node $n$ as $\text{ch}(n)$. Each node encodes a function, denoted as $h_n$ over a subset of input variables $\mathbf{X}_c \subseteq \mathbf{X}$, also called the *scope* of a node. A circuit has only three distinct computational nodes:

**Input node**  The *input node* or *leaf node* encodes an integrable parameterized function over input variables $\mathbf{X}_c \subseteq \mathbf{X}$. While the input node in probabilistic circuits encodes a probability distribution, in the case of DeepPCE it encodes a PCE, which we denote as $g$. Recalling Section 2, a PCE over scope $\mathbf{X}_c$ has the form

$$g(\mathbf{X}_c) = \sum_{\boldsymbol{\alpha} \in \mathcal{A}_c} w_{\boldsymbol{\alpha}} \boldsymbol{\Phi}_{\boldsymbol{\alpha}}(\mathbf{X}_c). \tag{9}$$

**Product node**  The *product node* represents a product $\prod_{c \in \text{ch}(n)} h_c(\mathbf{X}_c)$. The scope of the product node is the union of the scope of its input nodes, $\bigcup_{c \in \text{ch}(n)} \mathbf{X}_c$. For tractable inference, it is required that a circuit is *structured-decomposable*. A circuit is decomposable if all children of product units have *disjoint scopes*, so that $\mathbf{X}_c \cap \mathbf{X}_{c'} = \emptyset \ \forall c, c' \in \text{ch}(n)$. For structural decomposability, it is further required that all pairs of product nodes $n, m$ with same scope decompose the scope in the same way, so that $\mathbf{X}_{\text{ch}_i(n)} = \mathbf{X}_{\text{ch}_i(m)} \ \forall i \in \{1, \dots, C\}$, where $C$ is the number of children in $\text{ch}(n)$ and $\text{ch}(m)$.

**Sum node**  The *sum node* computes the sum $s(\mathbf{X}_c) = \sum_{c \in \text{ch}(n)} w_{n,c} h_c(\mathbf{X}_c)$, where $\mathbf{w}_n = [w_{n,1}, \dots, w_{n,C}]$ are the trainable parameters of the node and $C$ are the number of children of that node. All sum nodes in a circuit have to satisfy the

*Accepted for the 8$^{th}$ Workshop on Tractable Probabilistic Modeling at UAI  (TPM 2025).*

structural condition *smoothness* for tractable inference. A circuit is smooth if, for all sum nodes, the children of a sum node have *same scope*, so $\mathbf{X}_c = \mathbf{X}_{c'} \; \forall c, c' \in \text{ch}(n)$.

If a circuit satisfies the structural properties of *smoothness* and *decomposability*, statistical moments of $f(\mathbf{X})$ can be computed by decomposing the complex integral into smaller, easier to solve integrals based on the observation that integration interchanges with summation and integration interchanges with decomposable multiplication [Choi et al., 2020]. Essentially, the integral gets 'pushed down' to the input nodes of the circuit, which can be evaluated easily. It is then sufficient to perform a forward pass through the circuit using the evaluated integrals at the circuit leaves to compute the desired statistical moment of the whole circuit $f(\mathbf{X})$.

In tensorized circuit representations, multiple nodes are parallelized and grouped to circuit *layers*. Usually, a layer includes multiple nodes encoding the *same function* over the *same scope*, but each with a distinct set of parameters [Peharz et al., 2020a], increasing the number of parameters and expressivity of the circuit. The layerwise representation yields two types of product layers: (i) product layers that compute the *outer product* $\mathbf{A}_{N \times N} = \mathbf{u} \otimes \mathbf{v}$, where $\mathbf{u}$ is the vector of all $N$ outputs from children nodes with scope $\mathbf{X}_c$ and $\mathbf{v}$ is the vector of all $N$ outputs from children nodes with scope $\mathbf{X}_{c'}$, also referred to in the circuit literature as a *Kronecker product*, (ii) product layers computing the *Hadamard product* $\mathbf{a} = \mathbf{u} \odot \mathbf{v}$ [Loconte et al., 2024]. Aside from the PCE input layer computing the tensor products $\mathbf{\Phi}$, we use Hadamard product layers in the DeepPCE.

## A.2 EXPECTATIONS

First, we want to compute the expectation of the whole circuit $f(\mathbf{X}) = \mathbf{Y}$:

$$\mathbb{E}[\mathbf{Y}] = \int p(\mathbf{x}) f(\mathbf{x}) \mathrm{d}\mathbf{x} \tag{10}$$

We first compute the expectation at the input nodes of the circuit and show how we can pass them through product and sum nodes to get the result at the output.

### A.2.1 PCE input layer

We recall that, following from the orthogonality property of PCE, the expectation of a single input node PCE $g_n$ over scope $\mathbf{X}_c$ is

$$\mathbb{E}[g_n(\mathbf{X}_c)] = w_{n, \boldsymbol{\alpha}_0} \tag{11}$$

To compute the expectation for an input node, we thus simply have to set all $w_{n, \boldsymbol{\alpha}_i} = 0 \; \forall \, i > 0$.

### A.2.2 Product layer

As shown in section A.1, the DeepPCE is a *decomposable* circuit — inputs to product nodes always have disjoint scopes $\mathbf{X}_c \cap \mathbf{X}_{c'} = \emptyset$. The product $p_{n,m}$ of two inputs $g_n, g_m$ is

$$p_{n,m} = g_n(\mathbf{X}_c) g_m(\mathbf{X}_{c'}) \tag{12}$$

Because the circuit is decomposable and the input distribution $p(\mathbf{X})$ factorizes, we get that $g_n(\mathbf{X}_c) \perp\!\!\!\perp g_m(\mathbf{X}_{c'})$. We can thus decompose the expectation of the product:

$$\begin{aligned} \mathbb{E}[p_{n,m}] &= \mathbb{E}[g_n(\mathbf{X}_c) g_m(\mathbf{X}_{c'})] \\ &= \mathbb{E}[g_n(\mathbf{X}_c)] \mathbb{E}[g_m(\mathbf{X}_{c'})] \qquad g_n(\mathbf{X}_c) \perp\!\!\!\perp g_m(\mathbf{X}_{c'}) \end{aligned} \tag{13}$$

We only have to pass the expectations computed in A.2.1 to the product node to compute its expectation.

### A.2.3 Sum layer

Recall that the circuit satisfies the structural property of *smoothness* — sums are always computed over inputs with same scope. In case of a outer product layer (sec. A.1), a sum node with index $k$ computes the sum

$$s_k = \sum_{n,m} w_{k,n,m} p_{n,m} \tag{14}$$

The expectation is then

$$\mathbb{E}[s_k] = \sum_{n,m} w_{k,n,m} \mathbb{E}[p_{n,m}] \tag{15}$$

We know the expectation of the product $\mathbb{E}[p_{n,m}]$ from A.2.2. As deeper circuits are only sequences of sum and product nodes, it suffices to compute the expectations at the input nodes and perform a forward pass through the circuit to compute $\mathbb{E}[\mathbf{Y}]$.

## A.3 COVARIANCES

Next we want to compute the covariance of all DeepPCE outputs $\mathrm{cov}(\mathbf{Y}, \mathbf{Y})$, which is a $O \times O$ positive definite matrix. We show how covariances are computed at the input layer can be propagated through the network.

### A.3.1 PCE input layer

First, we compute the covariances at the input nodes, which are:

$$\mathrm{cov}(g_n(\mathbf{X}_c), g_m(\mathbf{X}_{c'})) = \mathbb{E}[g_n(\mathbf{X}_c)\, g_m(\mathbf{X}_{c'})] - \mathbb{E}[g_n(\mathbf{X}_{c'})]\mathbb{E}[g_m(\mathbf{X}_{c'})] \tag{16}$$

Recall that we assume a factorized distribution of the inputs $\mathbf{X}$. The covariance for all pairs of input node PCEs $g_n(\mathbf{X}_c), g_m(\mathbf{X}_{c'})$ with disjoint scopes $\mathbf{X}_c \neq \mathbf{X}_{c'}$ is thus zero, so we only have to compute the covariances from input nodes that share the same scope. We know how to compute $\mathbb{E}[g_n(\mathbf{X}_c)]$ and $\mathbb{E}[g_m(\mathbf{X}_c)]$ from (11). For a PCE input node, we can compute the second moment, i.e. the expectation of a product of two PCEs $g_n, g_m$, over the same scope as

$$
\begin{aligned}
\mathbb{E}[g_n(\mathbf{X}_c)\, g_m(\mathbf{X}_c)] &= \mathbb{E}\left[ \sum_{\boldsymbol{\alpha} \in \mathcal{A}_c} \sum_{\boldsymbol{\beta} \in \mathcal{A}_c} w_{n,\boldsymbol{\alpha}} w_{m,\boldsymbol{\beta}} \boldsymbol{\Phi}_{\boldsymbol{\alpha}}(\mathbf{X}_c) \boldsymbol{\Phi}_{\boldsymbol{\beta}}(\mathbf{X}_c) \right] \\
&= \sum_{\boldsymbol{\alpha},\boldsymbol{\beta} \in \mathcal{A}_c} w_{n,\boldsymbol{\alpha}} w_{m,\boldsymbol{\beta}} \underbrace{\mathbb{E}[\boldsymbol{\Phi}_{\boldsymbol{\alpha}}(\mathbf{X}_{c,\mathcal{I}}) \boldsymbol{\Phi}_{\boldsymbol{\beta}}(\mathbf{X}_{c,\mathcal{I}})]}_{=\delta_{\boldsymbol{\alpha},\boldsymbol{\beta}}}
\end{aligned} \tag{17}
$$

where $\boldsymbol{\alpha}$ is the multi-index of the PCE $g_n$ and $\boldsymbol{\beta}$ is the multi-index of the PCE $g_m$. We know again from the orthogonality property that $\mathbb{E}[\boldsymbol{\Phi}_{\boldsymbol{\alpha}}(\mathbf{X})\boldsymbol{\Phi}_{\boldsymbol{\beta}}(\mathbf{X})] = \delta_{\boldsymbol{\alpha},\boldsymbol{\beta}}$, where $\delta_{\boldsymbol{\alpha},\boldsymbol{\beta}}$ is the Kronecker delta. The second moment is thus

$$\mathbb{E}[g_n(\mathbf{X}_c)\, g_m(\mathbf{X}_c)] = \sum_{\boldsymbol{\alpha} \in \mathcal{A}_c} w_{n,\boldsymbol{\alpha}} w_{m,\boldsymbol{\alpha}} \tag{18}$$

We get the covariance using the result from (11):

$$\text{cov}(g_n(\mathbf{X}_c), g_m(\mathbf{X}_c)) = \mathbb{E}[g_n(\mathbf{X}_c)g_m(\mathbf{X}_c)] - \mathbb{E}[g_n(\mathbf{X}_c)]\mathbb{E}[g_m(\mathbf{X}_c)]$$

$$= \sum_{\boldsymbol{\alpha} \in \mathcal{A}_c} w_{n,\boldsymbol{\alpha}} w_{m,\boldsymbol{\alpha}} - w_{n,\boldsymbol{\alpha}_0} w_{m,\boldsymbol{\alpha}_0}$$

$$= \sum_{\boldsymbol{\alpha} \in \mathcal{A}_c \setminus \{\boldsymbol{\alpha}_0\}} \sum_{\boldsymbol{\beta} \in \mathcal{A}_c \setminus \{\boldsymbol{\beta}_0\}} w_{n,\boldsymbol{\alpha}} w_{m,\boldsymbol{\beta}} \tag{19}$$

### A.3.2 Product layer

To compute second moments at the product layer, we again use $g(\mathbf{X}_c) \perp\!\!\!\perp g(\mathbf{X}_{c'})$ due to decomposability:

$$\mathbb{E}[p_{n,m}p_{k,l}] = \mathbb{E}[g_n(\mathbf{X}_c)g_m(\mathbf{X}_{c'})g_k(\mathbf{X}_c)g_l(\mathbf{X}_{c'})]$$
$$= \mathbb{E}[g_n(\mathbf{X}_c)g_k(\mathbf{X}_c)]\mathbb{E}[g_m(\mathbf{X}_{c'})g_l(\mathbf{X}_{c'})] \tag{20}$$

We know how to compute the expectation of products over the same scope from (18). We can now compute the covariance

$$\text{cov}(p_{n,m}, p_{k,l}) = \mathbb{E}[p_{n,m}p_{k,l}] - \mathbb{E}[p_{n,m}]\mathbb{E}[p_{k,l}] \tag{21}$$

by applying (20) and (13). We are thus able to compute second moments at a product node by propagating second moments from the child nodes. Covariances can be computed similarly, with the additional need of computing expectations.

### A.3.3 Sum layer

Second moments of two sums $s_i, s_j$ over same scope are

$$\mathbb{E}[s_i s_j] = \mathbb{E}\left[ \sum_{n,m} w_{i,n,m} p_{n,m} \sum_{k,l} w_{j,k,l} p_{k,l} \right]$$
$$= \sum_{n,m,k,l} w_{i,n,m} w_{j,k,l} \mathbb{E}[p_{n,m}p_{k,l}] \tag{22}$$

We know how to compute the expectation of the product of two product nodes from (20). We finally are able to compute the covariance

$$\text{cov}(s_i, s_j) = \mathbb{E}[s_i s_j] - \mathbb{E}[s_i]\mathbb{E}[s_j] \tag{23}$$

using the result for the second moment of sums (22) and the expectation of sums (15). We show that statistical queries for the second moment $\mathbb{E}[\mathbf{Y}\mathbf{Y}]$ and the covariance $\text{cov}(\mathbf{Y}, \mathbf{Y})$ can be answered by computing those quantities at the input node PCEs. Leveraging the orthogonality property, they reduce to simple operations only involving the learned weights of the input nodes.

### A.4 CONDITIONAL EXPECTATIONS

Now, we consider the case of the conditional expectation $\mathbb{E}[\mathbf{Y}|\mathbf{x}_{\mathcal{I}}]$, where $\mathcal{I} \subseteq \{1, \ldots, D\}$ is an index set of some parameters in $\mathbf{X}$. The set of unconditioned variables is denoted as $\mathbf{X}_{\neg\mathcal{I}}$ with $\neg\mathcal{I} = \{1, \ldots, D\} \setminus \mathcal{I}$, so that $\mathbf{X}_{\mathcal{I}} \cup \mathbf{X}_{\neg\mathcal{I}} = \mathbf{X}$. The conditional expectation is

$$\mathbb{E}_{\mathbf{X}_{\neg\mathcal{I}}}[\mathbf{Y}|\mathbf{x}_{\mathcal{I}}] = \int p(\mathbf{x}_{\neg\mathcal{I}}|\mathbf{x}_{\mathcal{I}})f(\overbrace{\mathbf{x}_{\neg\mathcal{I}},\mathbf{x}_{\mathcal{I}}}^{\mathbf{x}})\mathrm{d}\mathbf{x}_{\neg\mathcal{I}}$$

$$= \int p(\mathbf{x}_{\neg\mathcal{I}})f(\mathbf{x}_{\neg\mathcal{I}},\mathbf{x}_{\mathcal{I}})\mathrm{d}\mathbf{x}_{\neg\mathcal{I}} \tag{24}$$

The second step again follows from the assumption that $p(\mathbf{X})$ factorizes, so that $p(\mathbf{x}_{\neg\mathcal{I}}|\mathbf{x}_{\mathcal{I}}) = p(\mathbf{x}_{\neg\mathcal{I}})$. We can write the conditional expectation of a PCE at the leaf layer as

$$\mathbb{E}_{\mathbf{X}_{c,\neg\mathcal{I}}}[g_n(\mathbf{X}_c)|\mathbf{x}_{c,\mathcal{I}}] = \mathbb{E}_{\mathbf{X}_{c,\neg\mathcal{I}}}\left[\sum_{\boldsymbol{\alpha}\in\mathcal{A}_c} w_{n,\boldsymbol{\alpha}}\boldsymbol{\Phi}_{\boldsymbol{\alpha}}(\mathbf{X}_{c,\neg\mathcal{I}})\hat{\boldsymbol{\Phi}}_{\boldsymbol{\alpha}}(\mathbf{x}_{c,\mathcal{I}})\right] \tag{25}$$

where $\hat{\boldsymbol{\Phi}}_{\boldsymbol{\alpha}}(\mathbf{x}_{c,\neg\mathcal{I}})$ is the tensor product of the conditional variables:

$$\hat{\boldsymbol{\Phi}}_{\boldsymbol{\alpha}}(\mathbf{x}_{c,\mathcal{I}}) = \prod_{x_e\in\mathbf{x}_{c,\mathcal{I}}} \phi_{\alpha_e}(x_e) \tag{26}$$

Based on (25), we get

$$\mathbb{E}_{\mathbf{X}_{c,\neg\mathcal{I}}}[g_n(\mathbf{X}_c)|\mathbf{x}_{c,\mathcal{I}}] = \sum_{\boldsymbol{\alpha}\in\mathcal{A}_c} w_{n,\boldsymbol{\alpha}}\hat{\boldsymbol{\Phi}}_{\boldsymbol{\alpha}}(\mathbf{x}_{c,\mathcal{I}})\underbrace{\mathbb{E}[\boldsymbol{\Phi}_{\boldsymbol{\alpha}}(\mathbf{X}_{c,\neg\mathcal{I}})]}_{\substack{=1\text{ if }\alpha_i=0\,\forall\,\alpha_i\in\boldsymbol{\alpha},\\ \text{else }0}}$$

$$= \sum_{\boldsymbol{\alpha}\in\mathcal{A}_{c,\mathcal{I}}} w_{n,\boldsymbol{\alpha}}\hat{\boldsymbol{\Phi}}_{\boldsymbol{\alpha}}(\mathbf{x}_{c,\mathcal{I}}) \quad\text{with }\mathcal{A}_{c,\mathcal{I}} = \{\boldsymbol{\alpha}:\ \forall j\in\neg\mathcal{I}:\alpha_j=0\} \tag{27}$$

It follows from the orthogonality property of the PCE that $\mathbb{E}[\boldsymbol{\Phi}_{\boldsymbol{\alpha}}(\mathbf{X}_{c,\neg\mathcal{I}})] = 1$ only if $\{\boldsymbol{\alpha}:\ \forall j\in\neg\mathcal{I}:\alpha_j=0\}$. To compute the conditional expectation, we set all leaf layer PCE weights to zero not meeting this requirement. Recycling our computations for expectations in product and sum layers (A.2), we only have to perform a forward pass to compute $\mathbb{E}_{\mathbf{X}_{c,\neg\mathcal{I}}}[\mathbf{Y}|\mathbf{x}_{\mathcal{I}}]$.

## A.5  CONDITIONAL COVARIANCES

We compute the conditial covariance $\mathrm{cov}_{\mathbf{X}_{\neg\mathcal{I}}}(\mathbf{Y},\mathbf{Y}|\mathbf{x}_{\mathcal{I}})$ using the same method as described in A.3. The PCE conditional second moment is

$$\mathbb{E}_{\mathbf{X}_{c,\neg\mathcal{I}}}[g_n(\mathbf{X}_{c,\neg\mathcal{I}}|\mathbf{x}_{\mathcal{I}})\,g_m(\mathbf{X}_{c,\neg\mathcal{I}}|\mathbf{x}_{\mathcal{I}})] =$$

$$= \mathbb{E}_{\mathbf{X}_{c,\neg\mathcal{I}}}\left[\sum_{\boldsymbol{\alpha},\boldsymbol{\beta}\in\mathcal{A}_c} w_{n,\boldsymbol{\alpha}}w_{m,\boldsymbol{\beta}}\hat{\boldsymbol{\Phi}}_{\boldsymbol{\alpha}}(\mathbf{x}_{\mathcal{I}})\hat{\boldsymbol{\Phi}}_{\boldsymbol{\beta}}(\mathbf{x}_{\mathcal{I}})\boldsymbol{\Phi}_{\boldsymbol{\alpha}}(\mathbf{X}_{c,\neg\mathcal{I}})\boldsymbol{\Phi}_{\boldsymbol{\beta}}(\mathbf{X}_{c,\neg\mathcal{I}})\right]$$

$$= \sum_{\boldsymbol{\alpha},\boldsymbol{\beta}\in\mathcal{A}_c} w_{n,\boldsymbol{\alpha}}w_{m,\boldsymbol{\beta}}\hat{\boldsymbol{\Phi}}_{\boldsymbol{\alpha}}(\mathbf{x}_{\mathcal{I}})\hat{\boldsymbol{\Phi}}_{\boldsymbol{\beta}}(\mathbf{x}_{\mathcal{I}})\underbrace{\mathbb{E}\left[\boldsymbol{\Phi}_{\boldsymbol{\alpha}}(\mathbf{X}_{c,\neg\mathcal{I}})\boldsymbol{\Phi}_{\boldsymbol{\beta}}(\mathbf{X}_{c,\neg\mathcal{I}})\right]}_{=\delta_{\boldsymbol{\alpha},\boldsymbol{\beta}}}$$

$$= \sum_{\boldsymbol{\alpha}\in\mathcal{A}_c\setminus\{\boldsymbol{\alpha}_0\}}\sum_{\boldsymbol{\beta}\in\mathcal{A}_c\setminus\{\boldsymbol{\beta}_0\}} w_{n,\boldsymbol{\alpha}}w_{m,\boldsymbol{\beta}}\hat{\boldsymbol{\Phi}}_{\boldsymbol{\alpha}}(\mathbf{x}_{\mathcal{I}})\hat{\boldsymbol{\Phi}}_{\boldsymbol{\beta}}(\mathbf{x}_{\mathcal{I}}) \tag{28}$$

We can now compute the covariance using (16). Based on A.3, we perform a forward pass to compute $\mathrm{cov}_{\mathbf{X}_{\neg\mathcal{I}}}(\mathbf{Y},\mathbf{Y}|\mathbf{x}_{\mathcal{I}})$.

## A.6 EXPECTATIONS OF COVARIANCES AND COVARIANCES OF EXPECTATIONS

Additional to computing statistical moments, we are interested in performing sensitivity analysis by identifying the input variables $X_d \in \mathbf{X}$ that contribute most to the variance of $f(\mathbf{X}) = \mathbf{Y}$ represented by the DeepPCE. First order Sobol indices are given by

$$S_i = \frac{\text{Var}_{X_i}(\mathbb{E}_{\mathbf{X}_{\neg i}}[\mathbf{Y} \mid X_i])}{\text{Var}(\mathbf{Y})}, \tag{29}$$

quantifying the contribution of *only* $X_i$ to the total variance of $\mathbf{Y}$. We thus are ultimately interested in computing the covariance of expectations conditioned on the random variables $\mathbf{X}_{\mathcal{I}} \subset \mathbf{X}$: $\text{cov}_{\mathbf{X}_{\mathcal{I}}}(\mathbb{E}_{\mathbf{X}_{\neg \mathcal{I}}}[\mathbf{Y}|\mathbf{X}_{\mathcal{I}}])$, quantifying the covariance based *only* on the randomness of $\mathbf{X}_{\mathcal{I}}$. We again apply the standard formula for the covariance

$$\begin{aligned}
\text{cov}_{\mathbf{X}_{\mathcal{I}}}(\mathbb{E}_{\mathbf{X}_{\neg \mathcal{I}}}[\mathbf{Y}|\mathbf{X}_{\mathcal{I}}], \mathbb{E}_{\mathbf{X}_{\neg \mathcal{I}}}[\mathbf{Y}|\mathbf{X}_{\mathcal{I}}]) &= \mathbb{E}_{\mathbf{X}_{\mathcal{I}}}[\mathbb{E}_{\mathbf{X}_{\neg \mathcal{I}}}[\mathbf{Y}|\mathbf{X}_{\mathcal{I}}] \, \mathbb{E}_{\mathbf{X}_{\neg \mathcal{I}}}[\mathbf{Y}|\mathbf{X}_{\mathcal{I}}]] - \mathbb{E}_{\mathbf{X}_{\mathcal{I}}}[\mathbb{E}_{\mathbf{X}_{\neg \mathcal{I}}}[\mathbf{Y}|\mathbf{X}_{\mathcal{I}}]]^2 \\
&= \mathbb{E}_{\mathbf{X}_{\mathcal{I}}}[\mathbb{E}_{\mathbf{X}_{\neg \mathcal{I}}}[\mathbf{Y}|\mathbf{X}_{\mathcal{I}}] \, \mathbb{E}_{\mathbf{X}_{\neg \mathcal{I}}}[\mathbf{Y}|\mathbf{X}_{\mathcal{I}}]] - \mathbb{E}[\mathbf{Y}]^2
\end{aligned} \tag{30}$$

For the simplification in the second step, we use the law of total expectation, which states that $\mathbb{E}[\mathbb{E}[Y|X]] = \mathbb{E}[Y]$. We know how to compute $\mathbb{E}[\mathbf{Y}]$ based on the derivation in A.2. We further apply our results from 27 to compute the second moment of the distribution of the conditional expectation at the PCE leaves:

$$\begin{aligned}
\mathbb{E}_{\mathbf{X}_{c,\mathcal{I}}}\Big[\mathbb{E}_{\mathbf{X}_{c,\neg \mathcal{I}}}[g_n(\mathbf{X}_{c,\neg \mathcal{I}})|\mathbf{X}_{c,\mathcal{I}}] \, \mathbb{E}_{\mathbf{X}_{c,\neg \mathcal{I}}}[g_m(\mathbf{X}_{c,\neg \mathcal{I}})|\mathbf{X}_{c,\mathcal{I}}]\Big] &= \\
&= \mathbb{E}\left[\sum_{\boldsymbol{\alpha} \in \mathcal{A}_{c,\mathcal{I}}} w_{n,\boldsymbol{\alpha}} \hat{\boldsymbol{\Phi}}_{\boldsymbol{\alpha}}(\mathbf{X}_{\mathcal{I}}) \sum_{\boldsymbol{\beta} \in \mathcal{A}_{c,\mathcal{I}}} w_{m,\boldsymbol{\beta}} \hat{\boldsymbol{\Phi}}_{\boldsymbol{\beta}}(\mathbf{X}_{\mathcal{I}})\right] \\
&= \sum_{\boldsymbol{\alpha},\boldsymbol{\beta} \in \mathcal{A}_{c,\mathcal{I}}} w_{n,\boldsymbol{\alpha}} w_{m,\boldsymbol{\beta}} \underbrace{\mathbb{E}\left[\hat{\boldsymbol{\Phi}}_{\boldsymbol{\alpha}}(\mathbf{X}_{\mathcal{I}}) \hat{\boldsymbol{\Phi}}_{\boldsymbol{\beta}}(\mathbf{X}_{\mathcal{I}})\right]}_{= \delta_{\boldsymbol{\alpha},\boldsymbol{\beta}}}
\end{aligned} \tag{31}$$

where $\mathcal{A}_{c,\mathcal{I}} = \{\boldsymbol{\alpha} : \forall j \in \neg \mathcal{I} : \alpha_j = 0\}$ as defined in (27). Based the orthogonality property of the PCE, the equation reduces to

$$\begin{aligned}
\mathbb{E}_{\mathbf{X}_{c,\mathcal{I}}}\Big[\mathbb{E}_{\mathbf{X}_{c,\neg \mathcal{I}}}[g_n(\mathbf{X}_{c,\neg \mathcal{I}}|\mathbf{X}_{c,\mathcal{I}})] \, \mathbb{E}_{\mathbf{X}_{c,\neg \mathcal{I}}}[g_m(\mathbf{X}_{c,\neg \mathcal{I}}|\mathbf{X}_{c,\mathcal{I}})]\Big] &= \\
&= \sum_{\boldsymbol{\alpha} \in \mathcal{A}_{c,\mathcal{I}}} w_{n,\boldsymbol{\alpha}} w_{m,\boldsymbol{\alpha}} \underbrace{\mathbb{E}\left[\hat{\boldsymbol{\Phi}}_{\boldsymbol{\alpha}}(\mathbf{X}_{\mathcal{I}}) \hat{\boldsymbol{\Phi}}_{\boldsymbol{\alpha}}(\mathbf{X}_{\mathcal{I}})\right]}_{= 1} \\
&= \sum_{\boldsymbol{\alpha} \in \mathcal{A}_{c,\mathcal{I}}} w_{n,\boldsymbol{\alpha}} w_{m,\boldsymbol{\alpha}}
\end{aligned} \tag{32}$$

We are now able to compute the covariance of conditional expectations at the PCE leaves using (34) by inserting (32):

$$\begin{aligned}
\text{cov}_{\mathbf{X}_{c,\mathcal{I}}}(\mathbb{E}_{\mathbf{X}_{c,\neg \mathcal{I}}}[g_n(\mathbf{X}_c)|\mathbf{X}_{c,\mathcal{I}}], \mathbb{E}_{\mathbf{X}_{c,\neg \mathcal{I}}}[g_n(\mathbf{X}_c)|\mathbf{X}_{c,\mathcal{I}}]) &= \sum_{\boldsymbol{\alpha} \in \mathcal{A}_{c,\mathcal{I}}} w_{n,\boldsymbol{\alpha}} w_{m,\boldsymbol{\alpha}} - w_{n,\boldsymbol{\alpha}_0} w_{m,\boldsymbol{\alpha}_0} \\
&= \sum_{\boldsymbol{\alpha} \in \mathcal{A}_{c,\mathcal{I}} \setminus \{\boldsymbol{\alpha}_0\}} w_{n,\boldsymbol{\alpha}} w_{m,\boldsymbol{\alpha}}
\end{aligned} \tag{33}$$

Finally, we are also able to compute the expectation of conditional covariances using the law of total covariance:

$$\mathbb{E}_{\mathbf{X}_{\mathcal{I}}}[\text{cov}_{\mathbf{X}_{\neg \mathcal{I}}}(\mathbf{Y}, \mathbf{Y}|\mathbf{X}_{\mathcal{I}})] = \text{cov}(\mathbf{Y}, \mathbf{Y}) + \text{cov}_{\mathbf{X}_{\mathcal{I}}}(\mathbb{E}_{\mathbf{X}_{\neg \mathcal{I}}}[\mathbf{Y}|\mathbf{X}_{\mathcal{I}}], \mathbb{E}_{\mathbf{X}_{\neg \mathcal{I}}}[\mathbf{Y}|\mathbf{X}_{\mathcal{I}}]). \tag{34}$$

by computing the covariance according to (16) and the covariance of conditional expectations according to (33).

# B MONTE CARLO EVALUATION

We test all inference queries by comparing the analytical results with Monte Carlo approximations of those quantities from the same DeepPCE. We compute Monte Carlo approximations with different sample sizes $S = [10^5, 10^6, 10^7, 10^8]$, performing 30 individual Monte Carlo runs per sample size setting. We then test the convergence with increasing sample size and compare it to the analytical solution derived earlier by performing one sample t-tests for all outputs, showing that the difference between the analytical solutions and the Monte Carlo approximations is not statistical significant for any outputs (Table 1). Figure 3 shows results for a DeepPCE with 8 inputs $\mathbf{X} \sim \mathcal{N}(\mathbf{0}, \mathbf{I})$ and 8 outputs, from which we randomly chose a single output for visualization. We iteratively validate the analytical solutions and use them in further Monte Carlo tests as described below.

Table 1: P-values computed using one sample t-tests, comparing the analytical solution of the statistical inferences to Monte Carlo approximations. The results are not statistical significant for any outputs, indicating that the analytical solution is not different from the Monte Carlo solution.

| Outputs | $\mathbb{E}[Y]$ | $\mathrm{Var}(Y)$ | $\mathbb{E}[Y|x]$ | $\mathrm{Var}(Y|x)$ | $\mathbb{E}[\mathrm{Var}(Y|X)]$ |
|---|---|---|---|---|---|
| $Y_1$ | 0.875 | 0.950 | 0.581 | 0.172 | 0.493 |
| $Y_2$ | 0.300 | 0.119 | 0.575 | 0.583 | 0.986 |
| $Y_3$ | 0.108 | 0.158 | 0.643 | 0.660 | 0.873 |
| $Y_4$ | 0.132 | 0.223 | 0.706 | 0.495 | 0.970 |
| $Y_5$ | 0.915 | 0.816 | 0.521 | 0.204 | 0.522 |
| $Y_6$ | 0.093 | 0.058 | 0.679 | 0.776 | 0.916 |
| $Y_7$ | 0.382 | 0.148 | 0.567 | 0.559 | 0.951 |
| $Y_8$ | 0.250 | 0.104 | 0.591 | 0.610 | 0.964 |

## B.1 EXPECTATION AND COVARIANCE

We first test the analytical computation of $\mathbb{E}[\mathbf{Y}]$ by comparing it with the Monte Carlo estimate

$$\mathbb{E}[\mathbf{Y}]_{MC} = \frac{1}{n} \sum_i^N \mathcal{M}(\mathbf{X}) \tag{35}$$

After confirming that $\mathbb{E}[\mathbf{Y}] = \mathbb{E}[\mathbf{Y}]_{MC}$, we apply the analytical expectation in the Monte Carlo approximation of the covariance:

$$\mathrm{cov}(\mathbf{Y}, \mathbf{Y})_{MC} = \frac{1}{n-1} \sum_i^N (\mathcal{M}(\mathbf{X}) - \underbrace{\mathbb{E}[\mathbf{Y}]}_{\text{tested}})^2 \tag{36}$$

## B.2 CONDITIONAL EXPECTATION AND CONDITIONAL COVARIANCE

For the validation of conditional moments, we condition on variables $\mathbf{X}_{\mathcal{I}} = \{X_1, X_2, X_3, X_5\}$. For tests with fixed conditional values, we chose conditioned values as $\mathbf{X}_{\mathcal{I}} \sim \mathcal{U}(-2, 2)$. We test the conditional expectation similar to (35):

$$\mathbb{E}[\mathbf{Y}|\mathbf{x}_{\mathcal{I}}]_{MC} = \frac{1}{n} \sum_i^N \mathcal{M}(\mathbf{X}_{\neg \mathcal{I}}, \mathbf{x}_{\mathcal{I}}) \tag{37}$$

After confirming the results, we use it for the Monte Carlo approximation of the conditional covariance

$$\mathrm{cov}(\mathbf{Y}, \mathbf{Y}|\mathbf{x}_{\mathcal{I}})_{MC} = \frac{1}{n-1} \sum_i^N (\mathcal{M}(\mathbf{X}_{\neg \mathcal{I}}, \mathbf{x}_{\mathcal{I}}) - \underbrace{\mathbb{E}[\mathbf{Y}|\mathbf{x}_{\mathcal{I}}]}_{\text{tested}})^2 \tag{38}$$

The Monte Carlo approximation for the expected conditional covariance is then simply

$$\mathbb{E}[\text{cov}(\mathbf{Y}, \mathbf{Y} | \mathbf{X}_{\mathcal{I}})]_{MC} = \frac{1}{n} \sum_i^N \underbrace{\text{cov}(\mathbf{Y}, \mathbf{Y} | \mathbf{x}_i)}_{\text{tested}} \tag{39}$$

Because we get the covariance of conditional expectations just from known quantities (34), we don't need to compare it to a Monte Carlo approximation.

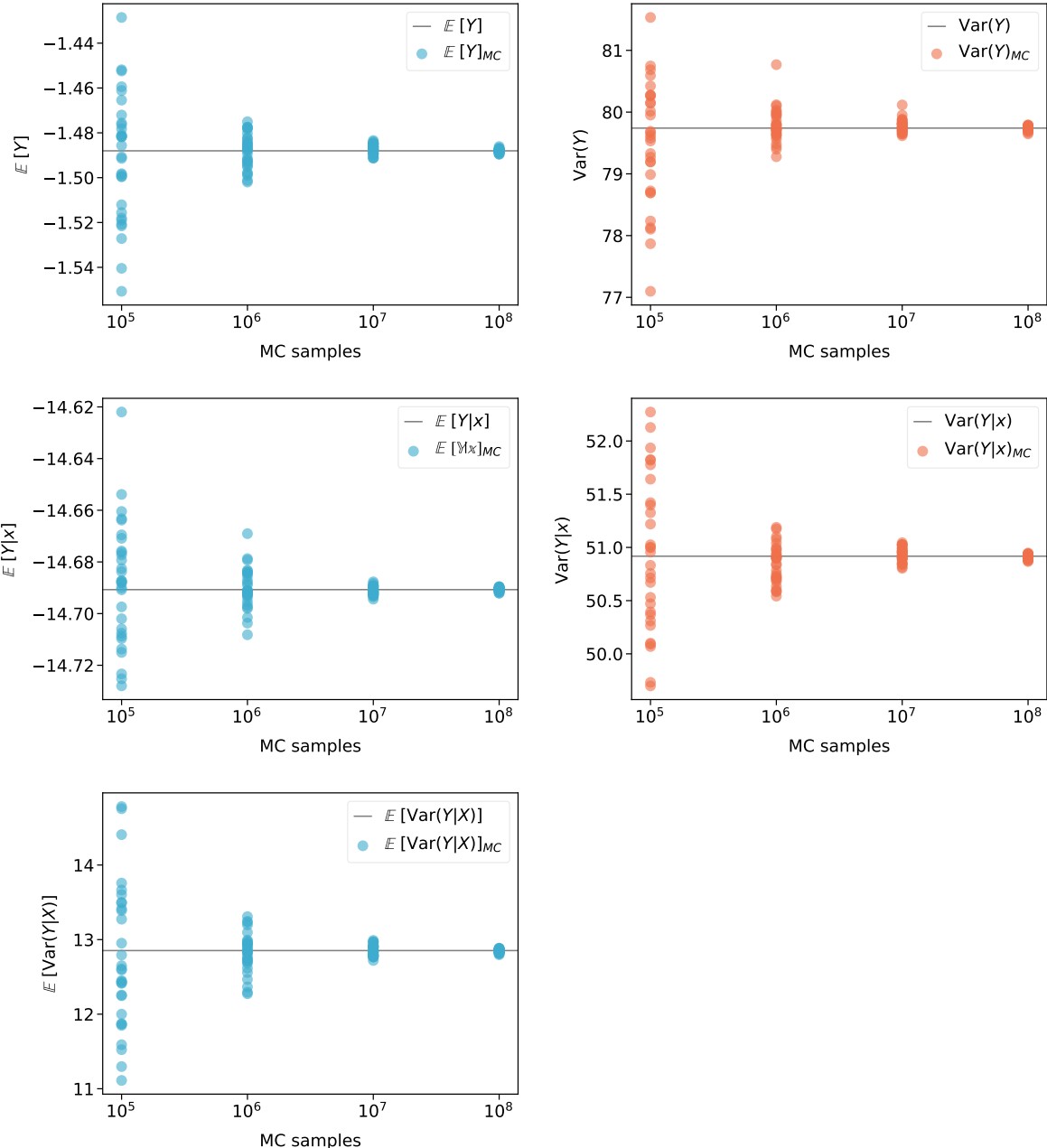

Figure 3: Inference pass evaluation comparing the analytical solution (grey line) to Monte Carlo approximations (blue/orange circles). The first row show unconditional expectation $\mathbb{E}[Y]$ (left) and unconditional variance $\text{Var}(Y)$, the second row conditional expectation $\mathbb{E}[Y|\mathbf{x}]$ and conditional variance $\text{Var}(Y|\mathbf{x})$, the third row shows the expectation of the conditional variance $\mathbb{E}[\text{Var}(Y|\mathbf{X}_{\mathcal{I}})]$

## C    BATCH NORMALIZATION AND ORTHOGONALITY

The DeepPCE formalization assumes that the input distribution factorizes, so that $p(\mathbf{X}) = \prod_d p(X_d)$. We show that orthogonality is preserved in the DeepPC also when batch normalization is applied at the sum layer. Batch normalization, denoted as the function $z$, applies an affine transformation at inference time:

$$z(\mathbf{Y}) = \gamma \frac{\mathbf{Y} - \mathbb{E}[\mathbf{Y}]}{\sqrt{\mathrm{Var}(\mathbf{Y}) + \varepsilon}} + \beta \tag{40}$$

where $\mathbf{Y}$ is previous hidden layer's output, $\gamma$ and $\beta$ are learnable parameters of the batch norm layer and $\mathbb{E}[\mathbf{Y}]$ and $\mathrm{Var}(\mathbf{Y})$ are the estimates for batch mean and variance respectively and $\varepsilon$ is a small constant added for numerical stability. At inference time, the batch statistics are independent from the inputs. The equation can be reformulated as

$$z(\mathbf{Y}) = \hat{\gamma}\mathbf{Y} + \hat{\beta} \tag{41}$$

$$\text{with } \hat{\gamma} = \frac{\gamma}{\sqrt{\mathrm{Var}[\mathbf{Y}] + \varepsilon}}, \quad \hat{\beta} = \beta - \frac{\gamma \mathbb{E}[\mathbf{Y}]}{\sqrt{\mathrm{Var}[\mathbf{Y}] + \varepsilon}} \tag{42}$$

Applied to a PCE $f(\mathbf{X}) = \sum_{\boldsymbol{\alpha} \in \mathcal{A}} w_{\boldsymbol{\alpha}} \boldsymbol{\Phi}_{\boldsymbol{\alpha}}(\mathbf{X})$ we get:

$$z(f(\mathbf{X})) = \hat{\gamma}\left( \sum_{\boldsymbol{\alpha} \in \mathcal{A}} w_{\boldsymbol{\alpha}} \boldsymbol{\Phi}_{\boldsymbol{\alpha}}(\mathbf{X}) \right) + \hat{\beta} \tag{43}$$

The batch norm coefficients $\gamma$ and $\beta$ can be subsumed in the weights and the zeroth-order term of the expansion, i.e. the constant $w_{\boldsymbol{\alpha}_0}$, respectively, thus not affecting orthogonality:

$$z(f(\mathbf{X})) = \sum_{\boldsymbol{\alpha} \in \mathcal{A}} \hat{\gamma}\, w_{\boldsymbol{\alpha}} \boldsymbol{\Phi}_{\boldsymbol{\alpha}}(\mathbf{X}) + b_{\boldsymbol{\alpha}} \quad \text{with } b_{\boldsymbol{\alpha}} = \begin{cases} \hat{\beta} & \text{if } \alpha_i = 0 \; \forall \alpha_i \in \boldsymbol{\alpha} \\ 0 & \text{otherwise} \end{cases} \tag{44}$$

## D    EXPERIMENT DETAILS

### D.1    TRAINING

Unlike classical PCE, DeepPCE does not admit a closed-form solution for the maximum likelihood estimate of the weights. Training therefore relies on standard MLP-style optimization, using gradient descent to minimize the $L_2$ loss. Because polynomial chaos expansions are computed at the DeepPCE input layer, the values at the leaf nodes can span several orders of magnitude, potentially destabilizing training. This issue is further amplified by the multiplicative interactions in product nodes, which, in high-dimensional settings, often lead to vanishing or exploding outputs. As a result, DeepPCE may converge to poor local minima or diverge entirely. This challenge is not unique to DeepPCE; training circuit-based models in high-dimensional regimes is known to be difficult with existing optimizers [Liu et al., 2022]. To mitigate the impact of large values introduced by higher-order polynomial terms, we draw inspiration from truncation strategies in the PCE literature [Blatman and Sudret, 2011]. Specifically, we initialize the weights of higher-degree polynomial terms with lower variances than those of lower-degree terms. We find that this variance scaling improves both training stability and convergence, but does not mitigate the problem entirely. Robust parameter initalization strategies for the DeepPCE are thus an important part of further research. We further apply batch normalization after each sum layer. Importantly, the learned batch normalization parameters can be absorbed into the constant (zeroth-degree) term of the polynomial expansion during inference, preserving both orthogonality and tractability of the inference pass (see Appendix C). For the experiments, we perform randomized training of multiple models, dropping runs with bad parameter sets.

### D.2    HYPERPARAMETERS

For each experiment, DeepPCE and MLP were tuned over the same set of hyperparameters (`learning rate`, `batch size`). Additionally, tuned parameters for the DeepPCE include the number of sum nodes per region (`num sums`),

the number of input variables per leaf scope (`scope size`) and the maximum order of the polynomial expansion (`max order`). Additional MLP tuning parameters include the number of hidden layers (`num hidden layers`) and the number of units per hidden layer (`num units`). We used early stopping to determine the end of training runs. All experiments were performed on a server with 252 GB RAM using a NVIDIA RTX A6000 GPU (48 GB Memory).

Table 2: Hyperparameters for the polynomial chaos benchmark experiment.

| DeepPCE | | MLP | |
|---|---|---|---|
| optimizer | adam | optimizer | adam |
| amsgrad | True | amsgrad | True |
| learning rate | 8.5e-3 | learning rate | 1.15e-3 |
| batch size | 16 | batch size | 128 |
| num sums | 40 | num hidden layers | 1 |
| scope size | 1 | num units | 3700 |
| max order | 3 | | |

Table 3: Hyperparameters for the Darcy flow experiment.

| DeepPCE | | MLP | |
|---|---|---|---|
| optimizer | adam | optimizer | adam |
| amsgrad | True | amsgrad | True |
| learning rate | 2.5e-3 | learning rate | 1.1e-3 |
| batch size | 16 | batch size | 64 |
| num sums | 25 | num hidden layers | 1 |
| scope size | 1 | num units | 600 |
| max order | 3 | | |

Table 4: Hyperparameters for the steady state diffusion experiment.

| DeepPCE | | MLP | |
|---|---|---|---|
| optimizer | adam | optimizer | adam |
| amsgrad | True | amsgrad | True |
| learning rate | 3.0e-2 | learning rate | 1.1e-3 |
| batch size | 16 | batch size | 128 |
| num sums | 30 | num hidden layers | 3 |
| scope size | 1 | num units | 3200 |
| max order | 3 | | |

## D.3   POLYNOMIAL CHAOS BENCHMARK

The benchmark 100D function [Lüthen et al., 2021] is defined as

$$f(\mathbf{X}) = 3 - \frac{5}{d}\sum_{i=1}^{d} iX_i + \frac{1}{d}\sum_{i=1}^{d} iX_i^3 + \frac{1}{3d}\sum_{i=1}^{d} i\ln(X_i^2 + X_i^4) + X_1X_2^2 + X_2X_4 - X_3X_5 + X_{51} + X_{50}X_{54}^2 \quad (45)$$

where inputs $\mathbf{X} = \{X_1, \dots X_D\}$ are sampled from uniform distributions $X_i \sim \mathcal{U}(1, 2) \ \forall i \neq 20$ and $X_{20} \sim \mathcal{U}(1, 3)$. Hyperparameters for the DeepPCE and the benchmark MLP are shown in 2. We compute additional Sobol indices for the DeepPCE, MLP, PCE(q=1) and PCE(q=0.8) models for different dataset sizes of $N = [1000, 5000, 50000]$ (Figure 4). It

can be observed that the performance of the DeepPCE is comparable to the PCE with $q = 0.8$, outperforming the shallow PCE variants at $N = 5000$. Using $N = 50000$, the DeepPCE and the PCE with $q = 0.8$ converge to a similar result. Results for PCE with $q = 1$ are not shown in this setting because they could not be computed with the available memory.

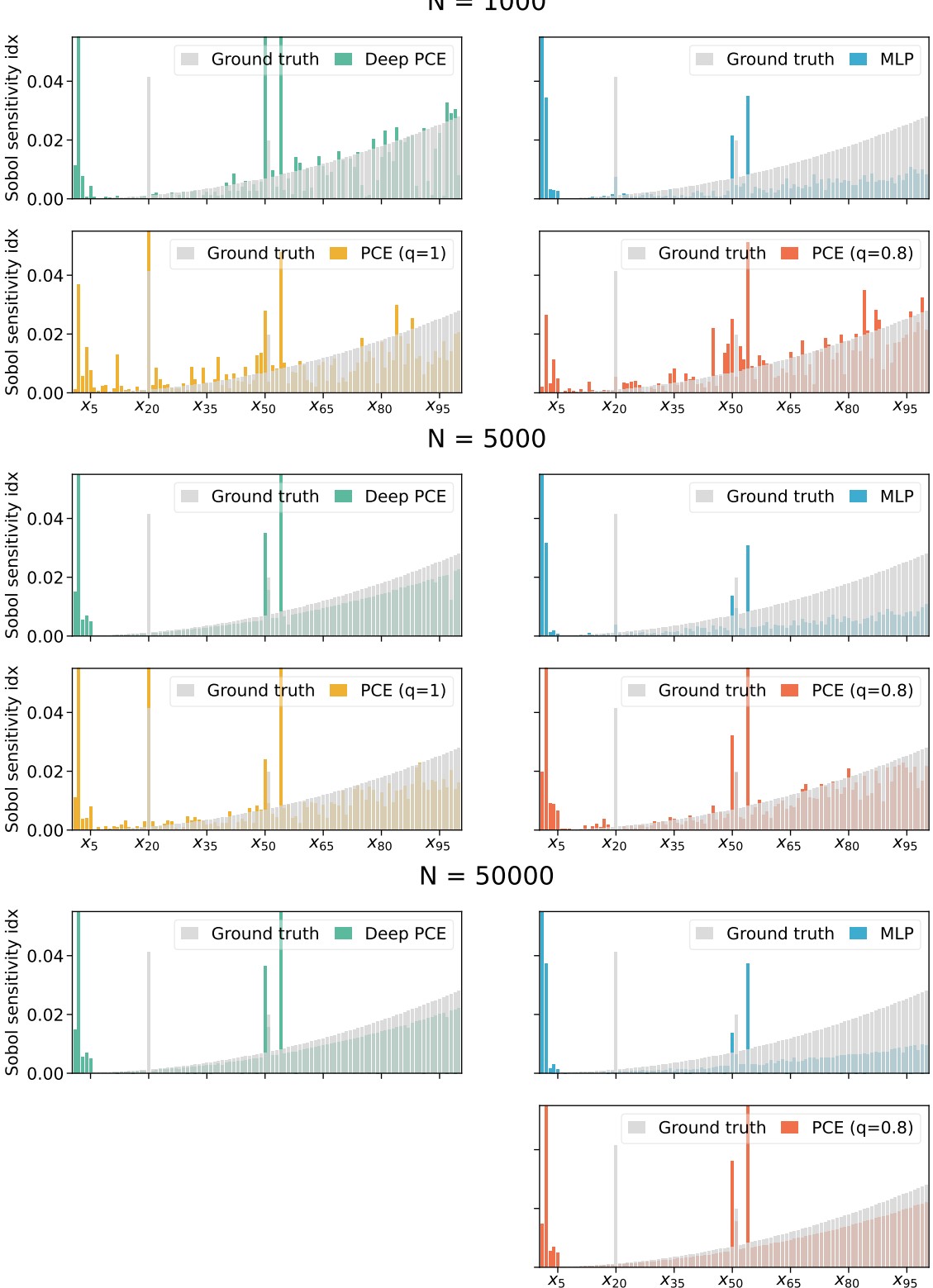

Figure 4: Sobol indices computed from models fitted with 1000, 5000 and 50000 training samples.

## D.4 DARCY FLOW

Hyperparameters for DeepPCE and MLP are shown in Table 3. Due to small absolute MSE values, we compute relative MSEs as

$$R = \frac{\frac{1}{N}\sum_i^N (y - \hat{y})^2}{\frac{1}{N}\sum_i^N y^2} \tag{46}$$

where $y$ are the true values and $\hat{y}$ are the model predictions. Relative MSEs for all test runs are shown in Table 5. Figure 5 shows random samples from the test set. In many cases, DeepPCE predictions are less noisy than MLP predictions, indicating the DeepPCEs applicability to approximate continuous problems such as PDEs. Train loss curves for the DeepPCE for all 20 runs are shown in Figure 6, showing that the DeepPCE is sensitive to certain parameter initializations.

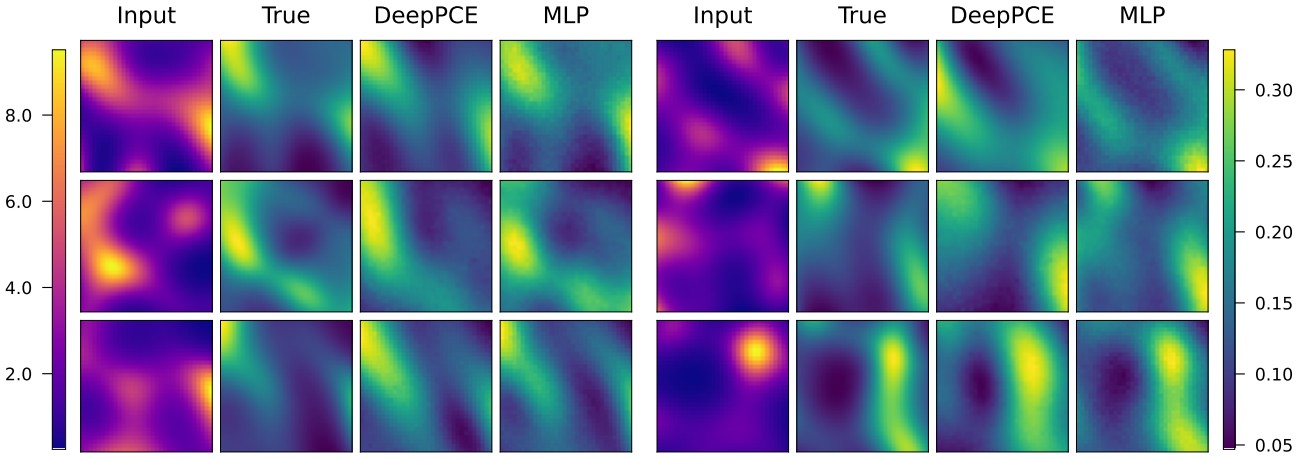

Figure 5: Random test samples for the **Darcy flow** experiment.

Table 5: Relative MSEs of DeepPCE and MLP on the Darcy flow dataset for all 20 test runs.

|  | run 1 | run 2 | run 3 | run 4 | run 5 | run 6 | run 7 | run 8 | run 9 | run 10 |
|---|---|---|---|---|---|---|---|---|---|---|
| DeepPCE | 0.0573 | 0.0688 | 0.2076 | 0.2075 | 0.1200 | 0.1423 | 0.0831 | 0.2077 | 0.0924 | 0.1676 |
| MLP | 0.0420 | 0.0444 | 0.0431 | 0.0437 | 0.0417 | 0.0425 | 0.0472 | 0.0476 | 0.0513 | 0.0441 |

|  | run 11 | run 12 | run 13 | run 14 | run 15 | run 16 | run 17 | run 18 | run 19 | run 20 |
|---|---|---|---|---|---|---|---|---|---|---|
| DeepPCE | 0.1088 | 0.0637 | 0.1052 | 0.0667 | 0.0644 | 0.0550 | 0.1747 | 0.0961 | 0.0523 | 0.0634 |
| MLP | 0.0428 | 0.0489 | 0.0429 | 0.0488 | 0.0457 | 0.0455 | 0.0453 | 0.0473 | 0.0479 | 0.0447 |

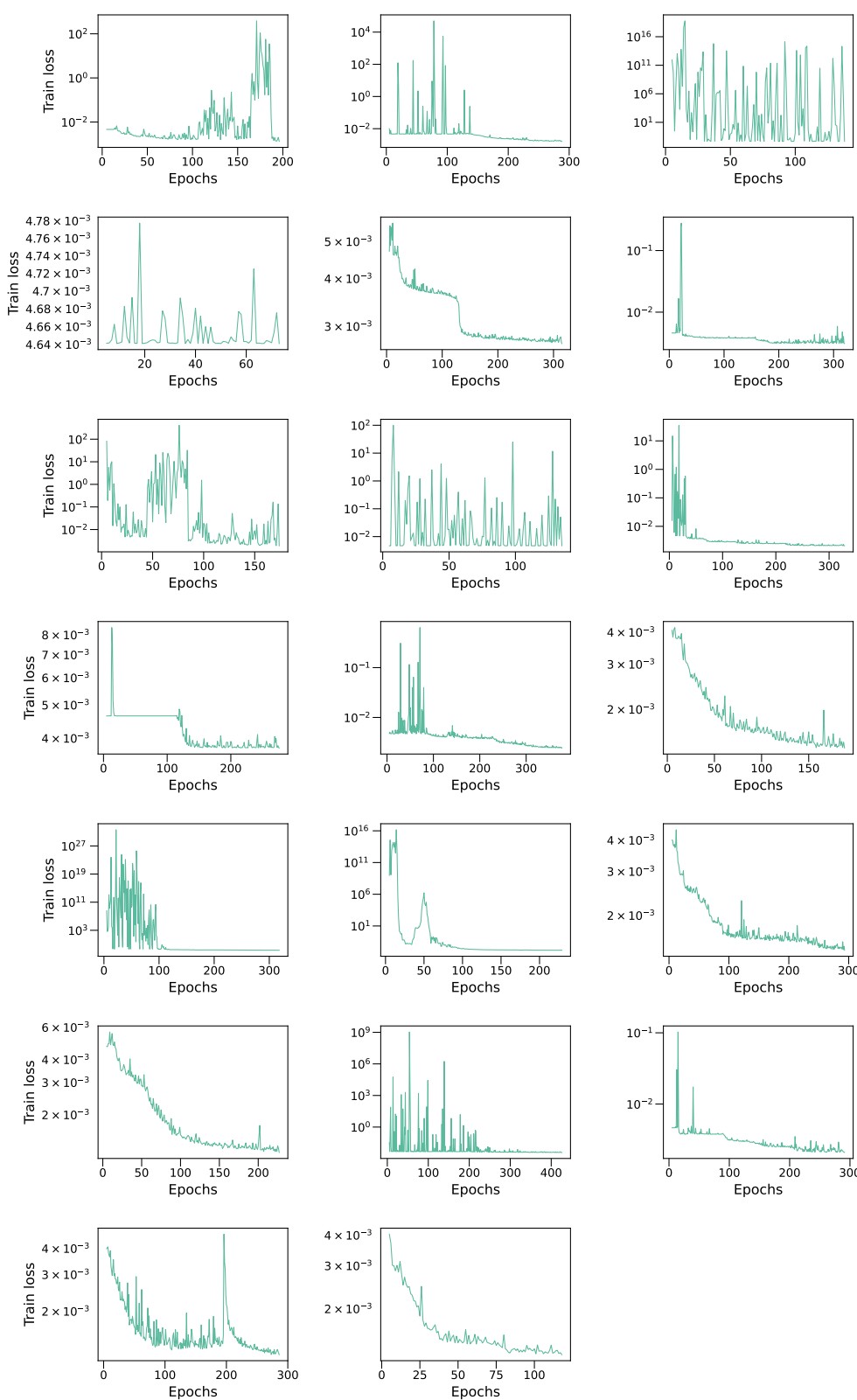

Figure 6: Train loss curves for all 20 runs of the DeepPCE for the Darcy flow experiment.

## D.5 STEADY STATE DIFFUSION

Hyperparameters for DeepPCE and MLP are shown in Table 4. We compute relative MSEs according to (46) for all runs, which are shown in Table 6. Random samples from the holdout test set are presented in Figure 7. DeepPCE train loss curves for all 20 runs are shown in Figure 8, showing that the DeepPCE is sensitive to certain parameter initializations.

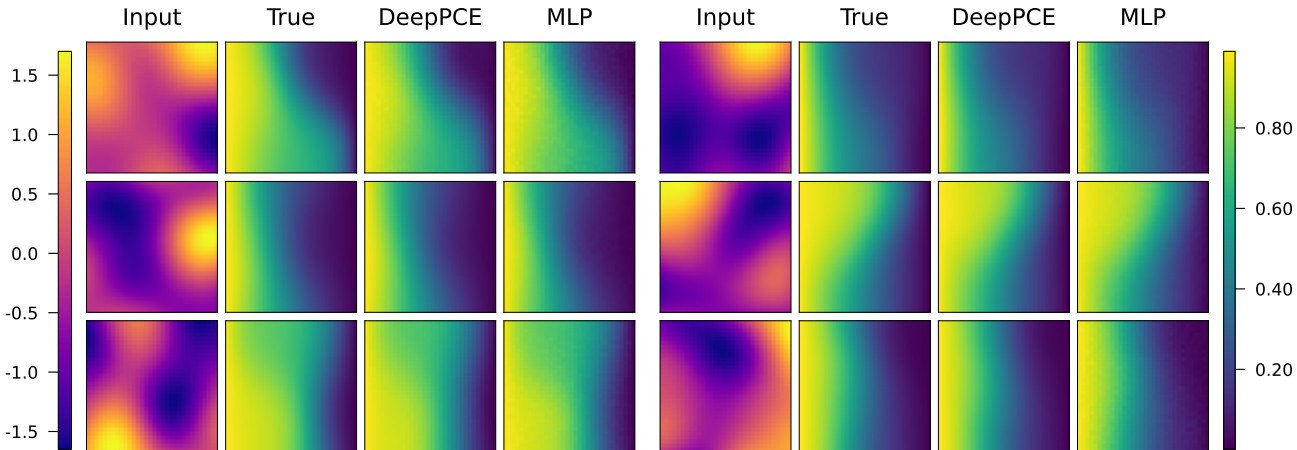

Figure 7: Random test samples for the **steady-state diffusion** experiment.

Table 6: Relative MSEs of DeepPCE and MLP on the steady state diffusion dataset for all 20 test runs.

|  | run 1 | run 2 | run 3 | run 4 | run 5 | run 6 | run 7 | run 8 | run 9 | run 10 |
|---|---|---|---|---|---|---|---|---|---|---|
| DeepPCE | 0.0005 | 0.0005 | 0.6792 | 0.0059 | 0.0004 | 0.0024 | 0.0001 | 0.0054 | 0.0038 | 0.0049 |
| MLP | 0.0003 | 0.0003 | 0.0003 | 0.0003 | 0.0003 | 0.0003 | 0.0003 | 0.0003 | 0.0003 | 0.0003 |

|  | run 11 | run 12 | run 13 | run 14 | run 15 | run 16 | run 17 | run 18 | run 19 | run 20 |
|---|---|---|---|---|---|---|---|---|---|---|
| DeepPCE | 0.0510 | 0.0032 | 0.0050 | 0.0028 | 0.0005 | 0.0087 | 0.0513 | 0.0047 | 0.0002 | 0.0003 |
| MLP | 0.0003 | 0.0003 | 0.0004 | 0.0003 | 0.0003 | 0.0003 | 0.0003 | 0.0003 | 0.0003 | 0.0003 |

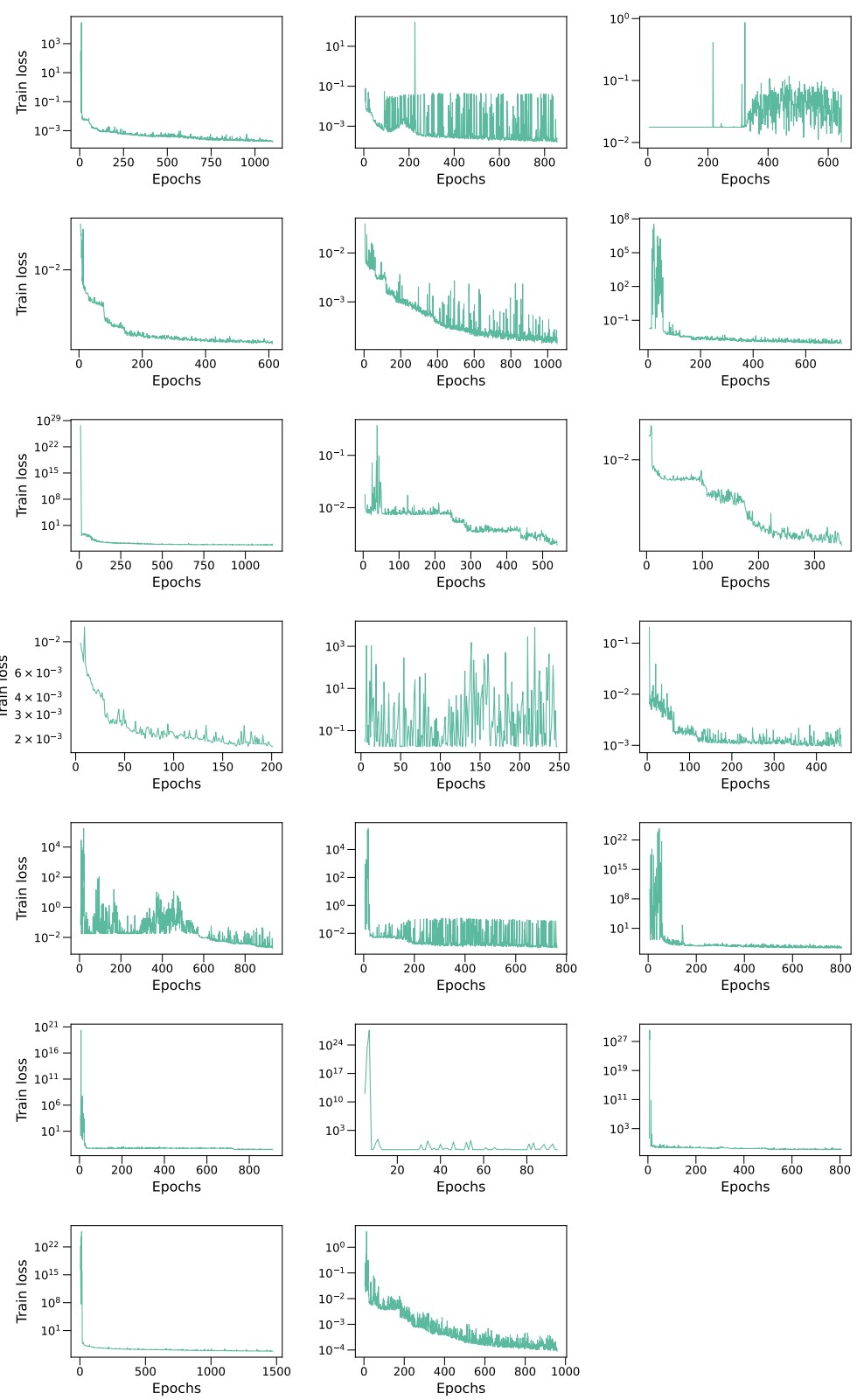

Figure 8: Train loss curves for all 20 runs of the DeepPCE for the steady state diffusion experiment.