# OpenReview forum: "Deep Polynomial Chaos Expansion"
_auai.org/UAI/2025/Workshop/TPM — TPM 2025_

### Official Review · Reviewer_WBQP · 2025-06-14
**Nice application of tractable models in the new domain**

**Rating:** 3

**Review:**

As I understood, the method is mostly about applying ideas from tractable probabilistic circuits in the domain of physics using different base. I like the connection to the base and that the requirement on tractability are similar. It would be actually nice to write the algebraic conditions on leafs under which the resulting computational graph is tractable.

I would be interested, how the method would work in practice on real problems. What would be the real gains.

---

### Official Review · Reviewer_YERr · 2025-06-16
**Interesting novel PC architecture with basis polynomials**

**Rating:** 3

**Review:**

This paper draws an interesting connection between polynomial chaos expansion (PCE) and probabilistic circuits (PC). In particular, it is proposed to represent high-dimensional functions as a PC with multivariate input nodes given by PCEs (basis expansions). This exploits the expressivity of PCEs along with the tractability and scalability induced by decomposable & smooth PCs (multilinear polynomials). The method proposed is sensible and principled, and the authors also tackle practical problems such as initialization to overcome the multiplicative gradient vanishing/exploding problem induced by the PC function. Overall, this is a good paper that pushes scalability of PCE and is highly relevant to TPM.

Comments and suggestions:
- What is the advantage of PCE relative to e.g. a decomposable and smooth PC (where expectation, variance, Sobol indices etc.) are also easy to compute)? My understanding is that it comes down to greater expressivity. If so, would it make sense to compare to a PC-only baseline with univariate inputs?
- There could be some more explanation on what role the distribution p(X) plays in the context of PCE. If this can be represented as a circuit then there exist approaches for computing the expectation of f(X) with respect to p e.g. [1].
- "The integrals for the expectation E[f(X)] and the variance Var(f(X)) in PCEs can be computed very efficiently even for high dimensional input spaces" - it's unclear what this means, as e.g. Eq (5) appears to scale exponentially in the dimension.
- Although the paper mentions smoothness and decomposability as the tractability conditions, it seems that computing covariance requires structured-decomposability (which seems to also be the condition mentioned at the end of the "PCE layer" paragraph.

[1] Khosravi et al. (2019). On tractable computation of expected predictions.

**Nominate For Best Paper:**

["Yes"]